# Primary material supply configurations and domestic recycling for cost-effective battery material production in the US

Jannis Wesselkaemper [1,2] ✉, Purabi Thakre[3], Alecia Ward[3] &
Andrew Z. Haddad [1] ✉

Battery cathode active material costs hinge on regionally concentrated, price-volatile metal supply. Here. we construct a regional facility-level cost model based on over 80 global lithium, cobalt, and nickel mines, refineries, and battery-grade material plants. Our model yields aggregated lithium, nickel, manganese, and cobalt production material costs from 392 region-based supply configurations for five different cathode active materials. Focusing on the United States, all-domestic supply is 9–34% costlier than global average, increasing by cobalt content, while these shortfalls can be overcome by selective low-cost material imports. Furthermore, we analyze costs of two U.S.-based recycling facilities from primary data and techno-economic modelling and compare resulting cathode active material-level costs to primary supply. Although it is still significantly higher on cathode active material cost-level, rising end-of-life flows and lowered black-mass prices will, however, make secondary supply cost-competitive to domestic and foreign primary supply cost floors. Facility-level benchmarks reveal targeted import, scaling, and production cost optimization as levers for a resilient, cost-effective U.S. battery-material supply chain.

Global demand for lithium-ion batteries (LIBs) is expected to increase from approximately 700 GWh in 2022 to around 4.7 TWh by 2030, driven primarily by electric vehicle (EV) adoption, stationary storage, and portable electronics[1]. Within each LIB cell, the cathode active material (CAM), typically layered oxides such as NMC ($LiNi_xMn_yCo_zO_2$), remains the dominant cost element, contributing 30–50% pack-level bill of materials[2], while NMC production accounts for up to 45% of battery production costs[3,4]. For NMC production, in turn, lithium, cobalt, and nickel make up to over 70% of costs[3]. Because CAM costs scale nearly linearly with the prices of these metals, recent market volatility now governs battery price trajectories just as manufacturing-scale learning rates plateau[5].

Upstream supply of crucial CAM precursors (metal sulfates) is highly concentrated. Indonesia and the Democratic Republic of Congo (DRC) currently provide 59% (2.2 megatons) (Mt) of global primary nickel and 76% (220 kilotons) (kt) of global primary cobalt, respectively[6]. Chile and Argentina supply 28% of global lithium from brine feedstocks, whereas Australia produces 37% of the world's lithium from spodumene deposits[6], most of which is refined into battery-grade lithium salts in China[7]. China's dominance is even starker downstream, refining around 75% of the world's cobalt and nickel intermediates and over 60% of lithium chemicals to battery-grade materials[8]. Such geographic concentration creates strategic and economic vulnerabilities for regions pursuing battery-supply security[8,9].

Policy initiatives, including the United States' Inflation Reduction Act, Europe's updated Battery Regulation and Critical Raw Materials Act, India's Faster Adoption and Manufacture of Hybrid and Electric Vehicles (FAME) Scheme, and Canada's Critical Mineral Strategy, seek

[1]Energy Storage and Distributed Resources Division, Lawrence Berkeley National Laboratory, Berkeley, CA, USA. [2]Energy Analysis and Environmental Impacts Division, Lawrence Berkeley National Laboratory, Berkeley, CA, USA. [3]Program Development Department, Lawrence Berkeley National Laboratory, Berkeley, CA, USA. ✉e-mail: JannisWesselkaemper@lbl.gov; azhaddad@lbl.gov

to reshore battery supply chains from mining to recycling[10–13]. Yet policymakers and industry still lack a transparent view of how fully domestic metal feedstocks would affect CAM production costs or competitiveness against global alternatives. Recent metal-price declines further complicate the economics of domestic projects, accentuating the need for rigorous, facility-level cost analyses.

Next to mining, recycling offers an increasingly complementary feedstock[14], as typical processes of mechanical processing of end-of-life (EoL) batteries into metal-rich black mass, followed by hydro-metallurgical extraction, can already recover over 95% of battery-grade lithium, nickel, cobalt, and manganese[15]. However, the delivered-cost advantage of recycled versus primary material remains uncertain, particularly because recyclers compete with entrenched global low-cost suppliers in volatile markets.

Most prior CAM-cost studies rely on top-down commodity indices or proprietary cost curves that obscure plant-level realities[3,16–18]. Here, we introduce a bottom-up model that fuses publicly reported data from over 80 operating and planned lithium, cobalt, and nickel mines, refineries, and conversion plants worldwide. By integrating mine-to-gate costs with refining, conversion, and CAM synthesis steps, we calculate levelized production costs that are sensitive to specific facility-level inputs and can be re-weighted for flexible regional supply mixes. Additionally, we use primary data of recycling facilities processing black mass to CAM precursor materials to further analyze CAM material costs from secondary feedstocks and cost optimization potentials with growing EoL battery material supply, and benchmark these to various primary supply scenarios.

Using the United States as a case study, we estimate aggregated material costs (in USD GWh$^{-1}$ CAM) for NMC 95, 811, 622, 532, and 111 based on domestic primary supply (for description of CAM acronyms, see "Methods"). We benchmark these against alternative global supply scenarios, including feedstocks from Australia, Africa, and South America, to gauge prospective domestic-based supply cost-competitiveness. We then quantify CAM production material costs achievable with domestic recycling and identify the cost optimization pathways required for secondary feedstocks to undercut primary

domestic or global supply. Collectively, the analysis maps a data-driven pathway toward secure, cost-effective, and regionally diversified material supply for CAM production.

## Results

### Levelized lithium, cobalt, and nickel production costs, global mining, refining, and battery-grade material production

We quantified levelized production costs for battery-grade primary lithium carbonate (or lithium hydroxide), cobalt sulfate, and nickel sulfate by analyzing primary company data from 79 mining and refining facilities that are either operational, under construction, or in development. Figure 1 maps the locations and life-of-mine (LOM) capacities of all facilities. Detailed information on all facilities is provided in Supplementary Table 2 and in Supplementary Data 1.

To capture the complete cost stack of battery-grade material production, mine and refinery data were augmented with levelized conversion costs for intermediate products (e.g., spodumene concentrate (5–6% Li$_2$O) or cobalt hydroxide) into battery-grade chemicals. Conversion costs draw on primary company data and complementary techno-economic cost modeling (see Supplementary Tables 5 and 6).

Figure 2 ranks the 79 facilities by levelized unit costs (USD t$^{-1}$) of lithium carbonate equivalents (LCE), cobalt (in cobalt sulfate), and nickel (in nickel sulfate) against cumulative LOM production capacity. For lithium, most North American and European projects occupy the upper cost quartile, whereas facilities in Australia, Africa, and South America cluster at the low-cost end. Cobalt supply is dominated by the DR Congo, whose weighted average production costs (cobalt as by-product in copper mining) of 14,588 USD t$^{-1}$ Co is 46–56% lower than comparable operations in Canada and Cuba, respectively. The lowest nickel production costs are identified in Papua New Guinea, while Africa, Australia, Canada, Indonesia, and USA fall within 11,387–12,540 USD t$^{-1}$ on average. Notwithstanding these regional trends, individual facility costs for all three metals vary substantially (see Table 1 for LOM weighted regional averages). Due to high data availability, we defined lithium facility regions on a continental level, except North America

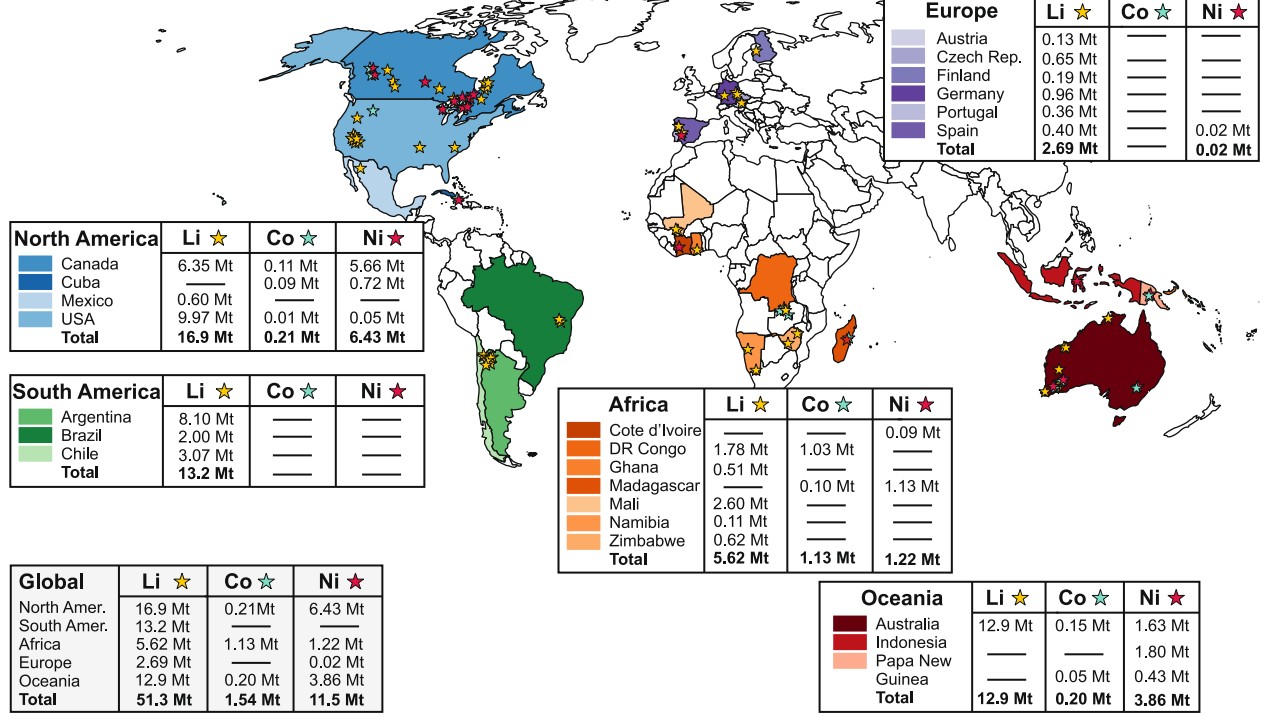

**Fig. 1 | Overview of location and life-of-mine (LOM) capacity of all analyzed material production facilities for lithium, cobalt, and nickel in this study.** Material production facilities are further grouped by nations and regions for regional supply scenario analysis in this study and reported in million tonnes (Mt).

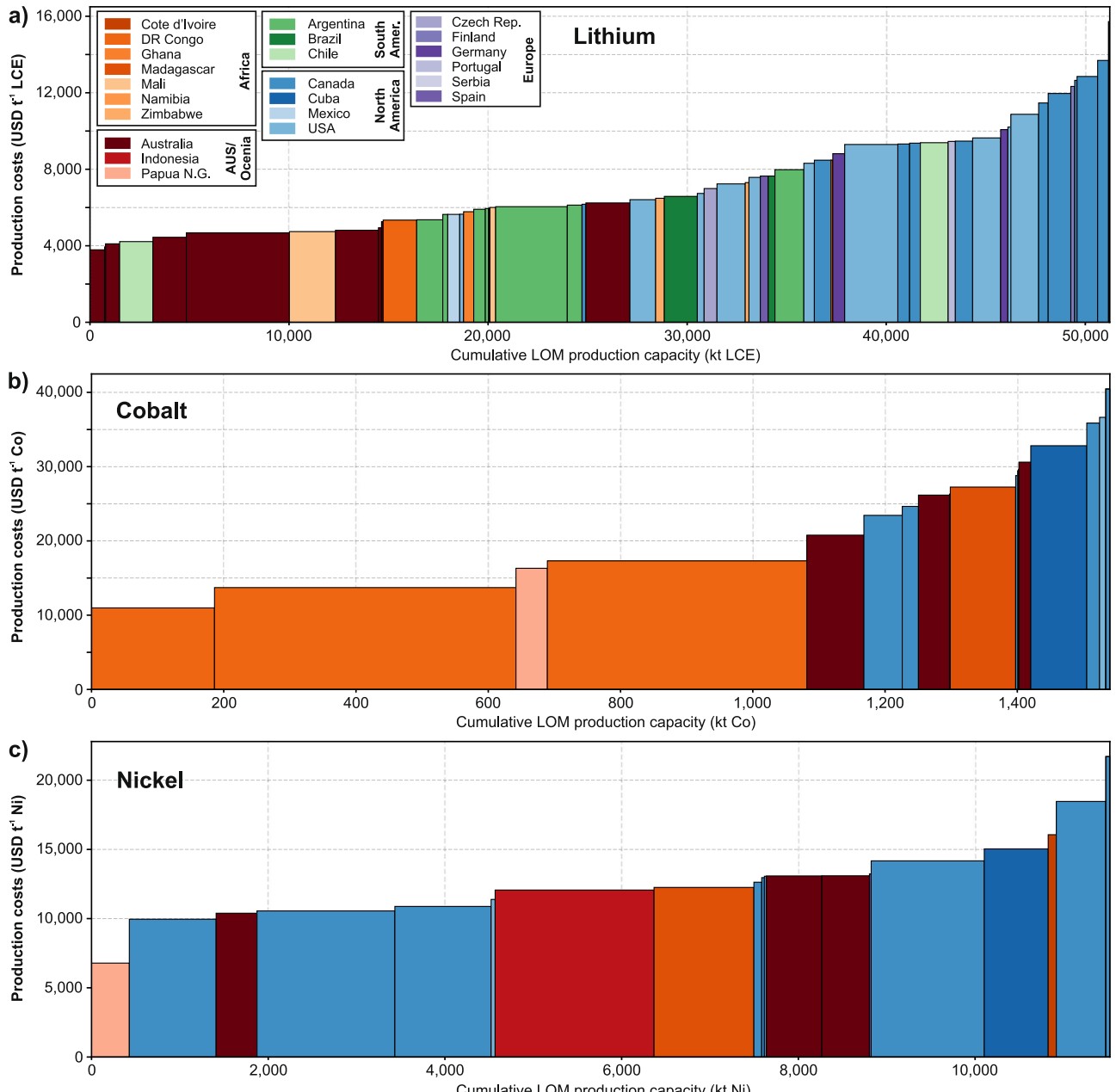

**Fig. 2 | Production costs of all analyzed material production facilities in this study: levelized material production cost (USD t⁻¹) vs. cumulative life-of-mine (LOM) supply capacity (kt).** a Lithium production costs in USD t⁻¹ lithium carbonate equivalents (LCE). **b** Cobalt production costs in USD t⁻¹ cobalt (as cobalt sulfate). **c** Nickel production costs in USD t⁻¹ nickel (as nickel sulfate). Facility production costs are presented in ascending order (project showing maximum lithium production costs is not presented: Wolfsberg Lithium Project (Austria) with $23,529 USD t LCE⁻¹ and 128.98 kilo-tonnes (kt) LCE LOM supply). All cost data entail levelized material production costs up to final products for battery cathode active material (CAM) production (Li₂CO₃/LiOH monohydrate; CoSO₄ heptahydrate; NiSO₄ hexahydrate). For detailed data on levelized conversion process costs for intermediate products exported from individual facilities (e.g., spodumene concentrate (5–6% Li₂O) from mines in Australia, or Co(OH)₂ from mines in DR Congo), see Supplementary Tables 5 and 6.

(separated for subsequent U.S. analysis). Cobalt and nickel averages are based on national levels (except Africa).

## Cost data aggregation, baseline primary sourced material costs of NMC cathode active material production

We aggregated regional levelized production costs for lithium, cobalt, and nickel to construct total material cost stacks for NMC CAMs. In addition to lithium carbonate (or lithium hydroxide), cobalt sulfate, and nickel sulfate, NMC synthesis consumes manganese sulfate, sodium hydroxide, and ammonium hydroxide. Levelized manganese sulfate production costs were derived from primary company cost data (see Supplementary Table 4). Because manganese, sodium

hydroxide, and ammonium hydroxide contribute only marginally to the total cost, unit costs are modeled as constant across all NMC formulations.

Figure 3 reports the resulting aggregated material costs (in million USD GWh CAM⁻¹) for NMC 95, 811, 622, 532, and 111. Baseline scenarios that source lithium, cobalt, and nickel entirely from U.S or European facilities are benchmarked against an average global supply mix, defined by 2024 regional production shares[6]. For example, lithium from Australia (36.7%), Chile (20.4%, etc.), or cobalt from the DRC (75.9%, etc.), and nickel from Indonesia (59.5%, etc.)[6]. Supplementary Tables 10–15 provide detailed information on the global supply scenario determination.

**Table 1 | Number of analyzed facilities and average lithium, cobalt, and nickel levelized production costs in regions as defined in this study**

| Material | Region (as defined in this study) | Number of facilities analyzed[a] | Average levelized production costs (weighted by LOM capacity) (USD t$^{-1}$) | By-products[b] |
|---|---|---|---|---|
| Li | Australia | 8 | 4632.3 | – |
| | Africa | 9 | 5384.4 | – |
| | Canada | 10 | 10,582.5 | – |
| | Europe | 7 | 9418.0 | – |
| | Mexico | 1 | 5641.7 | – |
| | South America | 11 | 6584.3 | – |
| | USA | 10 | 8603.0 | – |
| Co | Australia | 3 | 23,618.7 | Ni |
| | Canada | 7 | 27,011.3 | Cu, Ni, Au, Pt, Pd, Fe, Cr |
| | Cuba | 1 | 32,818.6 | Fe, Ni |
| | DR Congo | 3 | 14,588.3 | Cu |
| | Madagascar | 1 | 27,250.4 | Ni |
| | Papua New Guinea | 1 | 16,314.2 | Ni |
| | USA | 2 | 35,493.4 | Ni, Cu, Au |
| Ni | Australia | 3 | 12,321.35 | Co |
| | Africa | 2 | 12,540.70 | Cu, Co |
| | Canada | 9 | 12,261.59 | Cu, Co, Au, Pt, Pd, Fe, Cr |
| | Cuba | 1 | 15,037.14 | Fe, Co |
| | Indonesia | 1 | 12,065.21 | – |
| | Papua New Guinea | 1 | 6784.91 | Co |
| | Spain/Europe | 1 | 13,228.74 | Cu |
| | USA | 1 | 11,387.34 | Cu, Co |

By-product cost shares are determined based on market value-based cost allocation (for more detail see Methods and Supplementary Table 3). All cost data entail levelized material production costs up to final products for battery cathode active material (CAM) production (Li$_2$CO$_3$/ LiOH monohydrate; CoSO$_4$ heptahydrate; NiSO$_4$ hexahydrate).
[a]Due to overlaps in nickel and cobalt production, the number of facilities analyzed in this table exceeds 79.
[b]By-products which were considered in the market value-based cost allocation methodology (see "Method" section).

Figure 3 shows that material costs rise with cobalt content, climbing from NMC 95 to 111, while NMC 532 and 622 have similar costs. Across the three supply configurations, the global supply scenario is the cheapest, with 11.64 million USD GWh CAM$^{-1}$ for NMC 95 and 14.03 million USD GWh CAM$^{-1}$ for NMC 111. For cobalt-lean chemistries, a fully U.S.-sourced pathway remains competitive (12.69 million USD GWh CAM$^{-1}$ for NMC 95) relative to the Europe-sourced scenario (14.14 million USD GWh CAM$^{-1}$ for NMC 95). As cobalt content increases, however, the high cost of domestic cobalt drives the U.S. scenario above both the global and European benchmarks. The results reveal a need to lower U.S. supply costs if domestic NMC production aims to be globally cost-competitive.

**Primary supply cost scenario analysis, regional supply configurations, and cost-competitiveness of U.S.-based material supply**

To investigate cost-reduction pathways for U.S.-sourced primary material supply CAM production, we generated 392 regional supply

chain configurations per NMC chemistry, combining seven lithium, seven cobalt, and eight nickel sourcing regions, drawn from our facility dataset (see Table 1). Figure 4 plots the resulting distributions of aggregated material costs. Both the U.S.-global cost gap and the overall cost spread narrow with falling cobalt content (Fig. 4a). The range between the most expensive and least expensive scenarios differs by 12.8 million USD GWh$^{-1}$ for NMC 111 to 8.6 million USD GWh$^{-1}$ for NMC 95 (Fig. 4b). Notably, 44% of all configurations (172 scenarios) exceed the fully U.S.-sourced costs for NMC 95. This percentage declines with increasing cobalt content, emphasizing the relative competitiveness of cobalt-lean chemistries under domestic constraints.

Most configurations are cheaper than the U.S. supply scenario for every NMC chemistry, implying that selective imports of lithium, cobalt, and nickel sourced from other regions could decrease aggregated costs. We therefore modeled hybrid U.S.-focused supply strategies in which one metal remains U.S.-sourced, while either one or both of the other two metals are imported. Figure 5 delineates the break-even and more cost-effective supply configurations relative to the global supply scenarios as a benchmark.

In cobalt-lean NMC 95, substituting foreign lithium alone can close the cost-competitiveness gap (Fig. 5a). For high-cobalt chemistries, such as NMC 111, keeping cobalt domestic is cost-prohibitive. No combination of imported lithium or nickel attains cost parity with the global benchmark for NMC 111, and other cobalt-rich chemistries, like NMC 532 and 622, show only a few cost-competitive scenarios (Fig. 5b). These outcomes identify that domestic lithium and cobalt costs, rather than nickel, as the chief barriers to global cost-competitiveness as several mixes remain cost-effective when nickel is fixed to U.S. supply (Fig. 5a). Substantial additional savings are possible if nickel supply costs are further reduced, for example, by sourcing from Papua New Guinea (Fig. 5c), representing a major single lever to bring NMC 95 and NMC 811 within reach of the global cost floor.

**Benchmarking U.S. secondary supply costs, material sourcing from recycling and upcycling processes**

Next to domestic and foreign primary sources, domestically recycled battery materials represent a third material feedstock for U.S NMC CAM production. We examined two commercial recycling routes, based on primary company data and complementary techno-economic modeling, each beginning with black mass recovered after mechanical pre-treatment. (1) A hydrometallurgical recycling process, which converts black mass into lithium carbonate, plus sulfate salts of cobalt, nickel, and manganese, and (2) a hydrometallurgical upcycling variant, which produces lithium hydroxide and ((Ni$_x$Mn$_y$Co$_z$)(OH)$_2$) precursor (pCAM) in a process that can be flexibly doped with additional metal sulfates[15], allowing chemistry adaptation (e.g., converting a NMC 111 black mass input into a NMC 811 pCAM output). Using full-lifetime plant capacities and cost structures, we calculate levelized production costs for lithium, cobalt, nickel, manganese, and pCAM, mirroring the methodology applied to our primary supply analysis.

Figure 6 benchmarks the aggregated costs of the two recycling routes against both the global and the U.S. primary supply. Across all NMC chemistries, the hydrometallurgical recycling pathway is the highest, showing 18.2 million USD GWh$^{-1}$ for NMC 95, followed by the hydrometallurgical upcycling route at 15.6 million USD GWh$^{-1}$ for NMC 95. As the cobalt content in NMC rises, these premiums shrink. The recycling options approach, but do not yet fall below, the cost of the primary U.S. supply scenario. Because cobalt dominates in the CAM cost-stack, recycling processes become progressively more attractive for high-cobalt chemistries compared to low-cobalt NMC chemistries, although neither route attains cost parity with primary domestic mining supply.

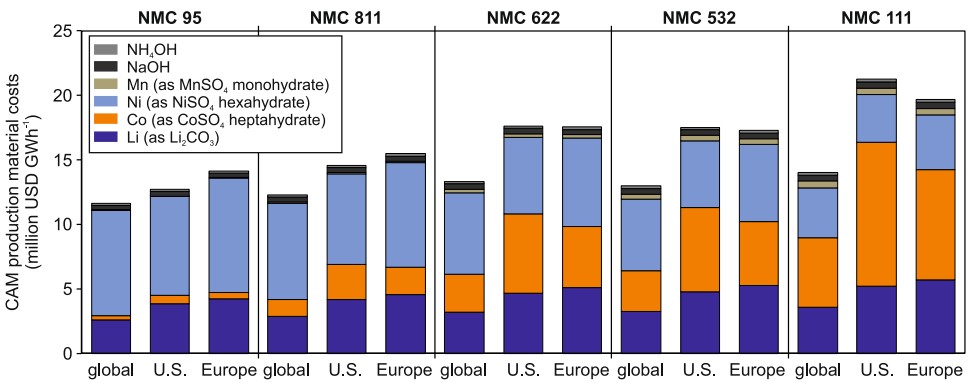

**Fig. 3 | Aggregated CAM production material costs\* for NMC 95, NMC 811, NMC 622, NMC 532, and NMC 111 in (1) 2024 global material supply scenario\*\*, (2) U.S.-sourced lithium, cobalt, and nickel scenario, and (3) Europe-sourced lithium, cobalt, and nickel scenario\*\*\* (in million USD GWh⁻¹).** \*Due to low-cost shares, costs per unit (USD t⁻¹) of manganese, sodium hydroxide, and ammonium hydroxide are constant across all NMC chemistries. \*\*Materials sourced according to regional material supply shares in 2024[6] for lithium (Australia 36.7%, Chile 20.4%, Zimbabwe 9.2%, Argentina 7.5%, Brazil 4.2%, Canada 1.8%, Portugal 0.2%, other 20.1%), cobalt (DR Congo 75.9%, Canada 1.6%, Australia 1.2%, Cuba 1.2%, other 20.1%), and nickel (Indonesia 59.5%, Canada 5.1%, Australia 3.0%, other 32.4%); due to missing data for some material production costs (e.g., lithium in China), all other countries are aggregated in 'other' (costs of 'other' are calculated based on the capacity-weighted average of global metal production costs of remaining mines analyzed in this study). \*\*\*Due to a lack of data for cobalt production costs in Europe, average Canadian cobalt production costs are assumed based on similar resource types, processes, and other economic factors (labor costs, etc.).

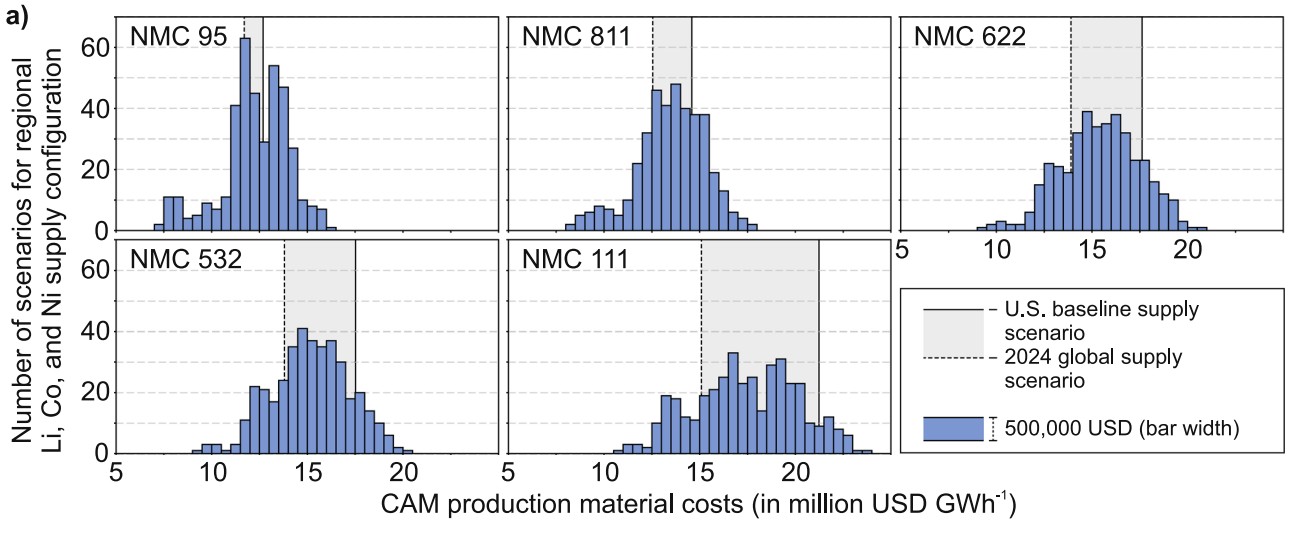

**Fig. 4 | Distribution of CAM production material costs (NMC 95, NMC 811, NMC 622, NMC 532, and NMC 111) for all regional lithium, cobalt, and nickel supply configurations, and comparison to aggregated costs of U.S. and global supply scenarios (in USD GWh⁻¹). a** Aggregated cost distribution for NMC chemistries: Absolute number of scenarios vs. aggregated costs of scenarios (grouped into 500,000 USD ranges). **b** Relative shares of the total number of scenarios above/under aggregated costs of U.S. and global supply scenario costs, including total aggregated costs of the lowest and highest cost scenario for each NMC chemistry.

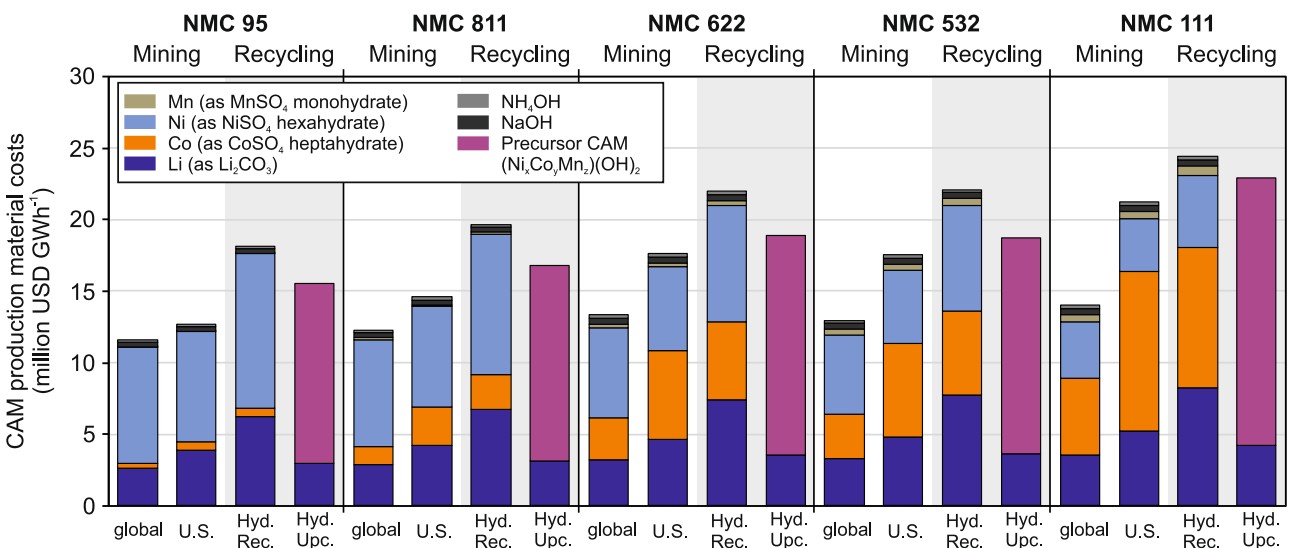

**Fig. 5 | U.S. supply configuration scenario analysis for optimization of CAM production material costs (for NMC 95, NMC 811, NMC 622, NMC 532, and NMC 111) and benchmarking to 2024 global supply scenario (in million USD GWh⁻¹).** **a** Variation of regional lithium and cobalt sourcing with fixed nickel supply from the USA. **b** Variation of regional lithium and nickel sourcing with fixed cobalt supply from the USA. **c** Variation of regional cobalt and nickel sourcing with fixed lithium supply from the USA. Compared benchmark costs of the 2024 global supply scenario vary between each NMC chemistry.

**Fig. 6 | Benchmarking of baseline CAM production material costs for materials sourced from hydrometallurgical recycling and hydrometallurgical upcycling for NMC 95, NMC 811, NMC 622, NMC 532, and NMC 111 (in million USD GWh⁻¹).** Hydrometallurgical recycling facility processes black mass to $Li_2CO_3$, $CoSO_4$ heptahydrate, $NiSO_4$ hexahydrate, and $MnSO_4$ monohydrate (input black mass metal contents equal output NMC chemistry); hydrometallurgical upcycling facility processes black mass to LiOH monohydrate and precursor CAM $(Ni_xMn_yCo_2)(OH)_2$ (pCAM) with metal content of output NMC chemistry (based on primary upcycling process data availability, model metal content of input black mass is approx. NMC 721 constantly across all output NMC chemistries; see also Supplementary Tables 17 and 19).

## Secondary supply cost scenario analysis, cost-competitiveness by black mass price reductions, and economies-of-scale

To gauge how secondary supply might reach cost-competitiveness with primary supplies, we modeled two optimization pathways grounded on recycling and upcycling cost structure. At the hydrometallurgical recycling facility, black mass accounts for 48% of total costs, followed by utilities (10%), and fixed operating expenditures (OpEx), such as labor (8%). In the hydrometallurgical upcycling route, black mass plus make-up metal sulfate costs contribute up to 70% of total costs, depending on metal shares in the input black mass and the targeted output pCAM. We therefore partition costs into feedstock costs, which account for black mass and supplemental metal sulfates, and all remaining items, which include utilities, labor, capital expenditures (CapEx), and others.

The first pathway considers reduced metal prices for lithium, cobalt, nickel, and manganese, which affect black mass and metal sulfate prices. We express black mass price as a function of inherent metal content (t), metal-specific payables (%), and prevailing metal prices (USD $t^{-1}$) (see also Methods and Supplementary Tables 20–22). Reducing those metal prices directly depresses both black mass and metal sulfate costs. The second pathway is based on economies-of-scale or learning effects of remaining costs (utilities, labor, CapEx, etc.) with increasing capacity. In the model, using Wright's Law, we applied a 10% learning rate to the remaining costs with a doubling of capacity, which is scaled according to a modeled EV end-of-life (EoL) NMC battery material supply in the United States (see Methods and Supplementary Table 23). Together, these scenarios quantify the levers by which recycling can approach, or fall below, primary supply cost benchmarks in both U.S. and global contexts.

Figure 7 quantifies these optimization levers. Figure 7a traces aggregate CAM material costs (USD $GWh^{-1}$ CAM) for both recycling pathways (hydrometallurgical recycling and hydrometallurgical upcycling) under two static feed-price scenarios, a moderate reduction and a minimum-price case, in which metal prices are floored at the lowest regional primary production costs among our analyzed mining facilities (e.g., 4632.3 USD $t$ $LCE^{-1}$ in Australia). These two static scenarios exclude in-time learning effects and represent aggregated CAM material costs optimization by lower metal prices (for more detail on metal price scenarios, see Supplementary Tables 21 and 22). Accounting for economies-of-scale impacting other cost drivers next to metal prices (e.g., CapEx and other OpEx), Fig. 7b, c shows learning effects based on baseline and minimum metal price scenarios, respectively, modeling three scenarios of EoL NMC battery supply feedstocks in the United States (basic, low, high).

At baseline black mass prices, scale-driven learning enables hydrometallurgical recycling to reach cost-parity with an all-U.S. primary supply, yet still sits above the global-mix and primary benchmarks. Upcycling displays a stronger chemistry dependence. The negligible residual-cost share in low cobalt NMC 95 limits learning benefits and leaves the route uncompetitive, whereas for NMC 622, larger residual costs allow economies-of-scale to push the pathway to break-even with the U.S. supply scenario over time. If metal prices fall, indirectly cutting both black-mass and make-up metal sulfate cost shares, domestic upcycling can rival, and in some instances undercut, the minimum global primary cost floor across all chemistries examined.

## Discussion

This study addresses the cost-analysis literature on NMC cathode materials by pairing a facility-level dataset with a techno-economic framework. Drawing on primary data from more than 80 mines, refineries, and conversion plants, each evaluated for LOM and levelized costs, our analysis sidesteps the opacity of commodity price indices and positions real production economics at the center of the cost stack. Because the model builds costs from the deposit type and flowsheet specifics, regional averages and single-site outliers become directly comparable, giving future academic studies and decision makers a clearer picture of where efficiencies or bottlenecks truly lie. We extend the same bottom-up methodology to two U.S. recycling routes, hydrometallurgical recycling, and hydrometallurgical upcycling, thereby enabling a direct comparison between mined and secondary feedstocks. Black mass costs are expressed as functions of embedded-metal value, payables, and market prices, while learning-curve effects are tied to projected EoL EV battery flows. Together, these price- and scale-driven levers quantify how falling metal prices and economies of scale can narrow the gap between recycled and primary supply. Finally, a combinatorial exploration of seven lithium, seven cobalt, and eight nickel sourcing regions (392 configurations per chemistry) maps cost-competitive hybrids that blend domestic resources with strategic imports, providing a template for the design of supply chain configurations.

Recent market behavior highlights the relevance of these insights. Spot cobalt has collapsed from more than 80,000 USD $t^{-1}$ in April 2022 to roughly 30,000 USD $t^{-1}$ in May 2025[19], while lithium carbonate has fallen from above 40,000 to around 10,000 USD t $LCE^{-1}$[20]. Our facility data reveal which producers remain profitable under such pressure and highlight a pipeline of projects that could extend the supply surplus by additional on-line-capacity, depressing prices further, and by extension, lowering black mass value, eventually. Under certain low black mass scenarios, U.S. hydrometallurgical recycling and upcycling edge close to imported low-cost primary supply, especially as domestic EoL battery volumes surge. A fully domestic lithium-cobalt-nickel is seldom cost-competitive, yet three levers emerge for the U.S.: selective imports that combine home-grown lithium or nickel with low-cost cobalt from the DR Congo or nickel from Papua New Guinea; equity stakes in foreign deposits, such as Albemarle's 49%-holdings in Greenbushes mine in Australia (remaining 51% owned by a joint venture of Chinese Tianqi Lithium Corporation and the Australian IGO Limited) and full ownership of the Salar de Atacama mine and La Negra refinery in Chile, which hedge against export bans (e.g., DR Congo's recent export stop of cobalt in February 2025[21]) and supply shocks[22]; and process innovation aimed at achieving Papua New Guinea-level nickel production costs domestically through reductions in reagents, capital expenditure, labor, and utilities. However, foreign investment activities have intensified in the recent decade, especially by Chinese companies[23], which challenges foreign investments in exploited mineral deposits, such as for cobalt in the DR Congo. For example, after the sale of U.S.-based Freeport-McMoRan's shares in the large Tenke Fungurume and Kinsanfu mines to China Molybdenum Company, the U.S. companies hold no ownership of cobalt deposits in DR Congo[24].

Achieving parity with Papua New Guinea's sub-7000 USD $t^{-1}$ Ni benchmark or the DRCs sub 15,000 USD $t^{-1}$ Co will require an integrated program of process innovation, without which, U.S CAM production will remain structurally high-cost even under the most favorable sourcing mixes identified in this study. Cost-effective process innovation, however, requires an in-depth understanding of production cost drivers, such as CapEx or OpEx (e.g., leaching and purification reagents, power, or labor)[25]. Therefore, priority research and development (R&D) avenues may include reducing leaching reagent intensities by either employing reagent-lean leaching chemistries such as glycine or recyclable solvents that cut acid and neutralization demand by significant percentages[25,26], developing entirely new extraction methodologies that forgo the use of high acid concentrations[27–29], or cost-effective recycling of sulfuric acid from current waste tailings[30]. Other strategies are needed to reduce front-end capital costs. This is beginning to be seen through modularization of high-pressure acid leaching (HPAL) and pressure oxidation (POX) units for the extraction of nickel and cobalt. Such modular, additively manufactured "plant-in-a-box" skids advancements promise cost

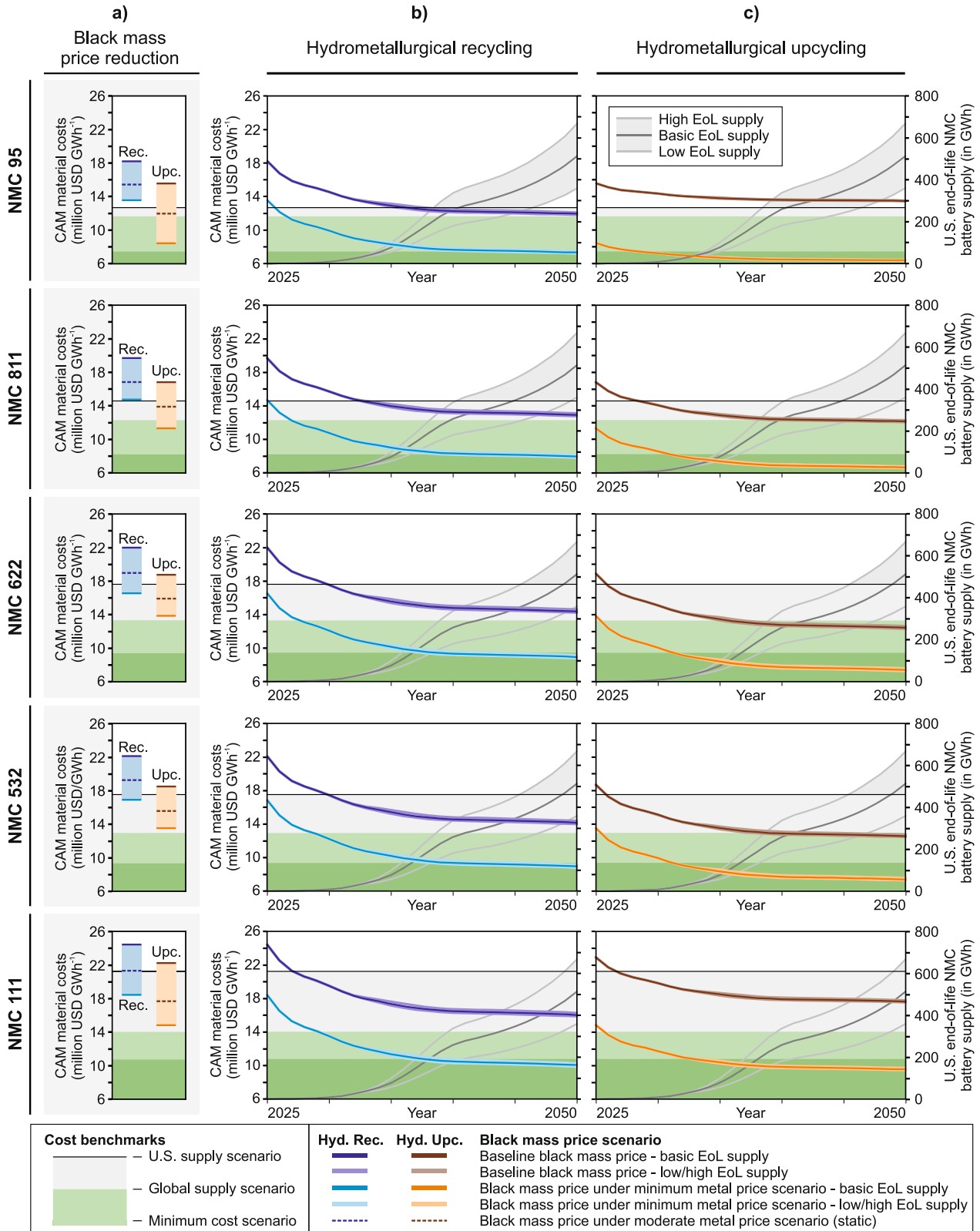

**Fig. 7 | Scenario analysis of aggregated cost optimization for material sourced from hydrometallurgical recycling and hydrometallurgical upcycling in the United States for NMC 95, NMC 811, NMC 622, NMC 532, and NMC 111 (in million USD GWh⁻¹). a** Static reduction of black mass price from baseline to moderate and minimum metal price scenario. **b** Learning effects of hydrometallurgical recycling with increasing end-of-life (EoL) NMC battery material supply in the United States for baseline and minimum metal price scenarios. **c** Learning effects of hydrometallurgical recycling with increasing end-of-life (EoL) NMC battery material supply in the United States for baseline and minimum metal price scenarios. Based on primary data for hydrometallurgical upcycling, metal content of input black mass is approx. NMC 721 constantly across all output NMC chemistries, thus, black mass, metal sulfate, and remaining cost (utilities, labor, capital expenditures, etc.) shares vary between output NMC CAM.

reduction through shortened construction schedules, improved maintenance schedules, and minimization of on-site labor[31]. Enhanced ore comminution strategies may also minimize site energy demands and unnecessary downstream throughput. For example, microwave-based ore processing has been validated at pilot scale to enhance mineral recovery and aid in preferential extraction, together reducing the comminution energy requirements by up to 24%, improving throughput by 30%, and minimizing tailings generation and water usage[32–34]. Together, these advances, as well as other emerging refinement R&D strategies, could close the current cost gap between domestic costs and the global floor. Absent of them, the analysis presented here indicates that even strategic import hybrids will not render U.S. or other fully domestic CAM primary material supply chains competitive at scale among most supply sourcing configurations, Fig. 5.

As a lever to potentially reducing supply risks by growing independence from import, recycling will play a larger role as EoL volumes grow[14]. Hydrometallurgical upcycling becomes particularly attractive for cobalt-rich chemistries, and, when metal prices fall further, can rival the minimum global primary-supply cost across all NMC grades. Policymakers could accelerate this crossover by incentivizing black mass collection, supporting plant scale-up, and funding R&D that lowers reagent and energy intensities of recycling processes.

The study's scope nonetheless leaves room for future work. Conversion capacity and cost data for cobalt and nickel sulfate plants are sparse and heavily China-centered[8]. Extending our framework to more primary data on these steps would improve the representativeness of regional cost curves further. In addition, despite the large coverage of most operating and planned material mining and refining facilities globally, public information on individual, high-capacity concessions, such as Chinese lithium and Indonesian nickel assets, remains limited. Thus, incorporating cost data for these operations would further enrich the data set by important assets and sharpen both the global benchmark and import-substitution strategies. Finally, while our analysis optimizes supply costs GWh⁻¹ of CAM, it does not model a dynamic U.S. NMC demand. Present production costs and supply curves and current and future demand trajectories globally and in the United States influence the need for more mining and refining capacity and, in turn, metal spot prices depending on production costs of add-on capacity globally or in individual, low-cost regions. Future research should therefore link cost and supply data to demand to investigate different geographic scenarios for feasible supply costs. Moreover, integrating other key inputs such as graphite, copper, cell and pack manufacturing costs, as well as transportation costs (which are not included here), and further regulatory parameters (such as tariffs and export restrictions) would deliver an end-to-end cost roadmap for total battery production.

By exposing facility-level economics and mapping optimal supply blends, this work equips industry and government with actionable benchmarks for investment, trade, and research priorities. Primary mining will remain indispensable in the near term[35], but strategic import partnerships and scaled domestic recycling offer credible pathways toward a resilient, cost-competitive U.S. battery-materials ecosystem.

## Methods

Throughout the analysis in the study, we use the bill-of-materials required to produce different lithium nickel manganese cobalt oxide (LiNi$_x$Mn$_y$Co$_z$O$_2$) (NMC) cathode active materials (CAM) as a functional unit to compare aggregated material costs of different material supply scenarios based on U.S. and foreign primary material sources, as well as U.S. secondary sources. As the main data source, we use a set of primary data containing over 80

company reports for globally operating and planned mining, refining, and conversion facilities. Furthermore, we complement this data sample with primary company data and techno-economic modeling of battery-grade material conversion facilities to obtain total levelized costs for battery-grade primary material supply of lithium carbonate (or lithium hydroxide), cobalt sulfate, nickel sulfate, and manganese sulfate that make up regional supply scenarios for NMC CAM production. Next, again, we leverage primary data complemented by techno-economic modeling to compare aggregated CAM production material costs for material supply sourced from U.S. recycling facilities for two processes (hydrometallurgical recycling and upcycling). Finally, a scenario analysis of recycling cost-optimization is based on black mass price modeling and an end-of-life (EoL) NMC battery supply model for the United States.

### Functional unit for cost aggregation, NMC cathode active material (CAM) production data

The bill-of-materials used in this study is sourced from Argonne National Laboratory's GREET model[2,36]. We constantly aggregate costs to material costs required for producing one GWh of NMC CAM for a 400-mile EV lithium-ion battery with either NMC 95, NMC 811, NMC 622, NMC 532, or NMC 111 as CAM[2,36]. Accordingly, NMC 811, for example, refers to CAM metal content ratios of 80% nickel, 10% cobalt, and 10% manganese, whereas NMC 95 represents an exception and refers to 95% nickel, 2.5% cobalt, and 2.5% manganese. Total CAM material demand (in t GWh⁻¹ CAM) is derived from GREET-model data on pack-level energy density (in kWh t⁻¹), pack-level CAM mass share (in wt% pack⁻¹), and CAM production material demand for lithium carbonate (or lithium hydroxide), cobalt, sulfate, nickel sulfate, manganese sulfate, as well as sodium hydroxide (NaOH) and ammonium hydroxide (NH$_4$OH) as process reagents, and precursor CAM (Ni$_x$Mn$_y$Co$_z$)(OH)$_2$) (pCAM) as an intermediate product (in t t⁻¹ NMC CAM). Demand for lithium, cobalt, and nickel metals is further obtained by molar mass-based recalculation of NMC production input materials. For all aggregated NMC CAM material costs in this study, unit costs for NaOH and NH$_4$OH are constant due to marginal cost shares (in USD t⁻¹). Detailed parameter values and bill-of-materials for CAM production of all NMC chemistries are listed in Supplementary Table 1.

### Levelized primary production costs by mining, refining, and conversion to battery-grade materials

To derive total levelized production costs ($LPC_{total}$) for lithium (as lithium carbonate equivalents (LCE)), cobalt (contained in cobalt sulfate), nickel (contained in nickel sulfate), and manganese (contained in manganese sulfate), we sum levelized production costs of mining ($LPC_{mining}$), refining ($LPC_{refining}$) and conversion ($LPC_{conversion}$) steps, according to Eq. (1):

$$LPC_{total}^i \left[\frac{USD}{t}\right] = LPC_{mining, j}^i \left[\frac{USD}{t}\right] + LPC_{refining, k}^i \left[\frac{USD}{t}\right] + LPC_{conversion, l}^i \left[\frac{USD}{t}\right] \tag{1}$$

where $i$ is the metal (lithium, cobalt, nickel, or manganese), and $j, k, l$ represent individual facilities. However, although the level of integration of material value-chain steps is facility-specific, the majority of mining facilities integrate following refining steps, whereas the conversion step is often performed in external facilities after shipping, especially for cobalt and nickel extraction facilities. In this study, however, transportation costs are not integrated into aggregated levelized production costs. Regardless of facilities' value-chain integration, the life-of-mine (LOM) levelized production costs of each facility ($LPC_m^i$) are systemically determined according

to Eqs. (2) and (3):

$$LPC_m^i \left[\frac{USD}{tLCE}\right] = \frac{Costs_{total, m}[USD]}{C_{LOM, m}^i[tLCE]} \tag{2}$$

$$\begin{aligned} Costs_{total, m}[USD] = &\ CapEx_{total, m}[USD] + OpEx_{total, m}[USD] \\ &+ SusEx_{total, m}[USD] + Costs_{closure, m}[USD] \end{aligned} \tag{3}$$

where $m$ is the respective facility, $Costs_{total, m}$ the total costs of the respective facility, $C_{LOM}^i$ the lifetime supply capacity of the metal $i$, $CapEx_{total, m}$ the total facility capital expenditures (CapEx), $OpEx_{total, m}$ the total facility operating expenditures (OpEx), $SusEx_{total, m}$ the total facility sustaining capital expenditures (SusEx), $Costs_{closure, m}$ and costs for facility closure or mine reclamation. Equations (2) and (3) represent facilities that produce a single metal, so that full facility costs are allocated to one product. However, caused by deposit type and inherent material value of deposits, almost all cobalt and nickel facilities (and some lithium facilities) produce by-products, such as copper, gold, cobalt, or nickel. In these cases, we apply a market value-based cost allocation approach[37] to determine costs of individual by-products, driven by metal prices and total-lifetime metal supply capacity (for more detail on by-product cost allocation methodology, see Supplementary Table 3).

Our primary data set comprises 79 company reports on mining and refining facilities for lithium, cobalt, nickel, and manganese. For conversion, we collected primary company data of four facilities for battery-grade lithium, cobalt, and nickel salt production. We complemented this conversion plant data set by market research data from S&P Global Market Intelligence (e.g., average LiOH monohydrate conversion costs from spodumene concentrate)[38], and techno-economic modeling of costs for additional conversion facilities (for details on conversion facility data and techno-economic modeling, see Supplementary Table 5). Similar to NaOH and NH₄OH, manganese sulfate only marginally contributes to the total NMC CAM cost. Thus, manganese unit costs (USD t⁻¹) are held constant at the lifetime-capacity-weighted average levelized production costs of analyzed manganese facilities in this study, which all integrate the production of battery-grade manganese sulfate (for details on manganese unit cost assessment, see Supplementary Table 4).

Based on the analysis of the facilities' levelized production costs for mining and refining, in the analysis of this study, we defined regional levelized material production costs by lifetime-capacity-weighted averages of facilities within these regions. For each facility in this region, we added average regional levelized conversion costs to the facility's levelized production costs for mining and refining (if required, depending on product type). Due to the large share of mining and refining primary company data in the data set, therefore, our regional supply chain configuration scenarios are mainly based on metal deposit locations, i.e., mining and refining activities, while we limit the scope of this study to assumptions of average regional conversion costs. Thus, in this study, we do not include a further layer of supply chain configurations by analyzing the cost-effective shipment to different conversion plants globally (for regional average conversion costs and assumptions on conversion supply chain configurations, see Supplementary Table 6).

For comprehensive, detailed information on ownership, location, deposit type, applied process, product type, by-products, lifetime, supply capacity, and cost structure of every analyzed facility in this study individually, as well as levelized production cost averages for regions as defined in this study, see Supplementary Data 1.

## Levelized secondary production costs by hydrometallurgical recycling and upcycling

In addition to levelized costs of primary material production facilities, we analyzed levelized material production costs for a hydrometallurgical recycling and a hydrometallurgical upcycling facility by, again, using primary company data and complementary techno-economic modeling (based on Argonne National Laboratory's EverBatt model[39] and literature[40]), according to Eqs. (2) and (3). Detailed levelized production costs NMC⁻¹ CAM chemistry are shown in the Supplementary Tables 16 and 17. The two analyzed recycling facilities process input black mass into output materials to produce NMC CAM.

The hydrometallurgical recycling plant produces lithium carbonate, cobalt sulfate, nickel sulfate, and manganese sulfate. Primary company data is based on a large-scale facility processing 35,000 tons year⁻¹ of black mass input with NMC 811 metal shares to a corresponding material output ratio on a metal basis (including losses by recovery rates). For the cost aggregation of other NMC CAMs, we recalculated input and output data towards remaining NMC 95, 622, 532, and 111 CAMs based on molar masses, so that input and output NMC shares are equal (for detailed primary and modeled input/output data of the hydrometallurgical recycling plant, see Supplementary Tables 16 and 18).

The hydrometallurgical upcycling facility converts input black mass into lithium hydroxide and target NMC pCAM by flexible doping with metal sulfate salts to adjust NMC chemistries. Primary company data is based on a facility processing 2000 tons year⁻¹ of approx. NMC 721 black mass into NMC 532 output NMC pCAM by doping with cobalt and manganese sulfate. In contrast to the hydrometallurgical recycling plant described above, in recalculations towards targeting the remaining CAM chemistries, we assumed a fixed black mass input across all output NMC CAM chemistries (approx. NMC 721), and adapted quantities of required dopants, as well as further reagents based on molar masses and techno-economic modeling (for detailed primary and modeled input/output data of the hydrometallurgical upcycling plant, see Supplementary Tables 17 and 19).

Because the primary company reports provide data mainly for CapEx (e.g., equipment costs or total cost-of-installation), reagent consumption, or utilities, we complemented these to total-lifetime facility costs by techno-economic modeling of other cost categories, such as labor or other OpEx (for more detail on recycling costs modeling, see Supplementary Tables 18 and 19).

Similar to mining and refining facilities extracting multiple metals, we determined levelized production costs for individual materials produced by the recycling processes (i.e., by-products) based on market value-based cost allocation (for more detail on cost allocation methodology, see Supplementary Table 3).

## Black mass price modeling

Key input material (or reagent) for hydrometallurgical recycling and upcycling processes is the black mass (containing lithium, cobalt, nickel, and manganese), whose costs are driven by black mass market prices, which, in turn, are dependent on inherent value. In this study, we model the black mass price ($BMP$; in USD t⁻¹) in the United States as a function of black mass metal contents ($M$; in t), metal payables ($P$; in %), and metal market prices ($Y$; in USD t⁻¹), according to Eq. (4):

$$BMP_{n,t}\left[\frac{USD}{t}\right] = \sum_i \left( M_i^n[t] * P_{i,t}[\%] * Y_{i,t}\left[\frac{USD}{t}\right] \right) \tag{4}$$

where $i$ is the metal (lithium, cobalt, nickel, or manganese), $n$ is the NMC chemistry of the black mass, and $t$ is the respective year. For the recycling cost scenario analysis in this study, we draw on data for recent black mass price and metal payable developments in North America provided by Benchmark Mineral Intelligence[41], and modeled the black mass price to obtain black mass price sensitivities and

developments in scenarios derived in this study. For details on black mass price calculation parameters and derived black mass prices under baseline and other metal price scenarios, see Supplementary Tables 20–22.

### Recycling cost learning effects (economies-of-scale) and EoL NMC battery supply modeling

In this study, modeled recycling cost optimizations are achieved by black mass price reductions and in-time learning effects of other recycling facility costs, such as CapEx, utility consumption, or labor costs. In academic literature, quantifications of learning effects (or economies-of-scale) are commonly based on Wright's Law, which determines cost reduction potentials by in-time learning effects of immature technologies with increasing market adoption and production capacity expansion[16]. In this study, to determine in-time cost optimization of recycling facilities, we model a cost floor[16] consistent of black mass and metal sulfate costs, and calculate cost reductions of remaining costs with growing production capacity. More specifically, we apply a learning rate of 10% for remaining cost optimization (CapEx, utilities, labor, etc.) each time process capacity doubles. We model the in-time capacity expansion of recycling facilities by using an EoL NMC battery supply forecast model for the United States based on prior literature[42]. In doing so, the model assumes that future U.S. black mass supply grows according to U.S. EoL NMC battery supply from EVs, which scales black mass processing capacity of hydrometallurgical recycling and upcycling facilities proportionally. For more details on EoL NMC battery supply modeling, see Supplementary Table 23. Equation (5) expresses Wright's Law as applied in this study for the total recycling facility costs:

$$Costs_{t,m}^{total}\left[\frac{USD}{t}\right] = \left(Costs_{0,m}^{total}\left[\frac{USD}{t}\right] - Costs_{floor,m}\left[\frac{USD}{t}\right]\right)* $$
$$\left(\frac{EoL_t^{NMC}[kWh]}{EoL_0^{NMC}[kWh]}\right)^{\log_2(1-r)} + Costs_{floor,m}\left[\frac{USD}{t}\right] \quad (5)$$

where $Costs_{t,m}^{total}$ are the total-lifetime recycling facility's costs at the time $t$, $Costs_{0,m}^{total}$ are the baseline total-lifetime recycling facility's costs, $EoL_t^{NMC}$ is the total EoL NMC battery supply in the United States at the time $t$, $EoL_0^{NMC}$ is the total baseline EoL NMC battery supply in the United States at the time $t = 0$ (year 2025 in this study), $Costs_{floor,m}$ is the cost floor consisting of black mass and metal sulfate salt costs, and $r$ is the learning rate.

## Data availability

The authors declare that the main data supporting the findings of this study are available within the article and its Supplementary Information files. All data sources and data presented in the main manuscript are included in Supplementary Data 1.

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

## Acknowledgements

Lawrence Berkeley National Laboratory is supported by the U.S. Department of Energy under Contract No. DE-AC02-05CH11231. This work was supported by the Department of Energy's Energy Efficiency and Renewable Energy (EERE) Office under project award number AWD00007865 and by LBNLs Program Development Office

## Author contributions

A.Z.H. conceived the idea, acquired funding and resources, supervised, performed investigation, formal analysis, visualization, and writing (original draft, review & editing). J.W. performed investigation, formal analysis, visualization, writing (original draft, review & editing). P.T. and A.W. acquired funding and contributed to writing review & editing).

## Competing interests

The authors declare no competing interests.
