## [Transparent Peer Review file · Nature Communications]

Primary material supply configurations and domestic recycling for cost-effective battery material production in the US

Corresponding Author: Dr Andrew Haddad

Version 0:

Reviewer comments:

Reviewer #1

(Remarks to the Author)

I have carefully reviewed the manuscript and find it to be a timely, well-executed, and highly valuable contribution to the field of battery materials and recycling economics. The paper addresses a significant and pressing challenge: understanding the cost parity between virgin and recycled materials across a wide array of cathode active materials (CAMs). This is particularly important in the context of growing demand for sustainable, circular, and economically viable battery value chains.

One of the manuscript's strongest features lies in its use of transparent, granular, and high-quality datasets for the production and recycling cost modeling. The authors provide detailed input assumptions across mining, refining, and hydrometallurgical recycling stages. This level of detail is rarely available in the literature and offers an essential foundation for both replication and future research. The regional differentiation, cost allocation methodology, and breakdown by chemistry enhance the practical value of the results.

Crucially, the inclusion of pricing and cost structures supported by public and semi-industrial sources opens the door to a broader use of this model and its assumptions. The modeling framework can be extended or adapted in future work to assess policy impacts, cross-regional competitiveness, or different recycling scenarios. The availability of structured and traceable price data fills a clear gap in the literature and can serve as a benchmark dataset for future techno-economic assessments in the battery space.

In addition, the paper is very well written. The flow of arguments is logical, the methodology is clearly explained, and the figures and supplementary materials are well designed and informative. The authors also make a clear effort to contextualize their results in terms of real-world relevance and future technology trajectories, particularly regarding the increasing role of recycled materials in sustainable battery production.

Although the study focuses on hydrometallurgical recycling, this is clearly stated and well justified. The exclusion of alternative technologies does not diminish the value of the work, given the depth of the analysis provided.

In summary, I strongly recommend this manuscript for publication. It offers excellent methodological rigor, high relevance to both academia and industry, and a uniquely valuable dataset that will undoubtedly serve as a reference point for ongoing and future studies in the field of battery recycling and materials sustainability.

Sincerely,
Sina Orangi

Reviewer #2

(Remarks to the Author)

This study takes a detailed approach to provide cost estimates for primary battery minerals (globally) and recycled minerals (U.S.). The study assesses various configurations of obtaining minerals supply that could reduce total battery costs for the U.S., making them more cost competitive. This is an impressive and detailed study, well-written, with interesting analysis and figures. The topic is highly relevant to the current policy landscape. I would be glad to see this study published in Nature Communications. A few comments to address are below.

Comments:

1. The effort you have taken to compile publicly available mine information is commendable. However, are you able to comment on the representativeness of this data, i.e. how well does publicly reported cost information cover the spectrum of costs associated with all mining projects, or those that are included in proprietary data?
2. Relatedly, in assessing the cost competitiveness of scenarios with different regional mixes, my question is to what extent we can assume that the average cost used for each region here is truly represent of average costs to meet much larger volume needs, e.g. to meet entire U.S. battery supply chain needs? I have this concern especially because it seems that the costs for several regions are based on only a single facility analyzed in a particular region (from Table 1), and as we see in Figure 2, there could be considerable variation in cost across regions. Moreover, costs may of course scale up over the course of a mine's lifetime, or change due to other volatilities in the market? To what extent is it a concern to here rely on a single average cost for each region, in making broader assessments about which strategies for regional sourcing could lower U.S. battery costs?
3. Figure 7 (and understanding all of the different cases) was a bit confusing for me and took a while to understand. The paragraph preceding this figure could benefit from more detailed description of the different scenarios here and the Figure itself could perhaps use better signposting of the different scenarios.

Minor comments:

1. Line 254-255: Wording ("surpass") is a little confusing here. The recycling options costs do "surpass" (are higher than) that of the U.S. primary supply scenario. Maybe change to "The recycling options approach, but do not yet fall below..."? Similar point in Line 279.

Reviewer #3

(Remarks to the Author)

The manuscript addresses the production costs of battery materials, particularly cathode active materials, from a U.S. perspective. It compares both primary and secondary material supplies using supply chain scenarios from different countries and regions. The calculations are based on a large sample of mining and processing facilities.

The manuscript covers an interesting and relevant topic. It is well-written and includes comprehensible and informative figures. The supplementary material supports a deeper understanding of the calculations and analyses performed. Some formal improvements are needed, mainly in the reference section. It would also be useful to discuss the impact of U.S. tariffs and their development on the results. Nevertheless, the paper is a valuable addition to the discussion on battery material supply.

Therefore, we recommend that the authors resubmit a revised manuscript that addresses the comments listed below (minor revision).

Major comments

- Throughout the paper, "NMC 95" is used instead of "NMC 955". If the combination of 90% nickel (Ni), 5% manganese (Mn), and 5% cobalt (Co) is meant, use the more common abbreviation "NMC 955". Otherwise explain what "NMC 95" means. This applies to many parts of the paper and the supplementary material.
- The abstract mentions "over 80" facilities. In line 79, for example, 79 facilities are mentioned. However, Table 1 and the supplementary Excel sheet list 93 facilities. The difference between 79 and 93 likely stems from facilities that process multiple materials. To increase clarity, the number of facilities examined should be explained once, and the resulting number (79 or 93) should be used throughout the paper and the abstract.
- The discussion section could be improved by stating the limitations of the calculations in the methods section (e.g., that transportation was not included in the results).
- As the paper focuses on the U.S., and many of the scenarios include material imports to the U.S., the discussion should address current U.S. tariffs and their development.
- Please carefully revise the reference section, as some references are incomplete (e.g., #23 has no author or year), contain spelling errors (e.g., #7 has "enrgy" instead of "energy"), or have strange abbreviations (e.g., #11 has "Commission, E." instead of "European Union"). Furthermore, different styles were used (e.g., #24 has no abbreviated first author), and the link in #20 does not work. The same applies to the reference section of the supplementary material.

Minor comments

- In line 144, a comma must be used instead of a period after the percentage sign in the parentheses after "DRC".
- In line 207, "as" should be replaced with "are" (the sixth word).
- In line 360, "modularity" is excessive.
- In the Methods section, some abbreviations are introduced again (e.g., NMC, CAM, EoL). This does not seem necessary.
- In line 432, "LOM" is referred to as "total-lifetime". On page 2, "LOM" was introduced as "life-of-mine". Please stick to one term.

Version 1:

Reviewer comments:

Reviewer #2

(Remarks to the Author)

Thanks, the authors have satisfactorily addressed my comments. I have no further comments.

Reviewer #3

(Remarks to the Author)

The authors have addressed all of my comments. In my opinion, the paper is ready for publication.

Response to reviewers' comments for:

Primary battery material supply configurations and domestic recycling costs—a U.S.-based landscape analysis for cost-effective battery production

We would like to thank the Editor and the Reviewers for taking the time to review our manuscript and for providing constructive comments and suggestions that have greatly helped us further improving the paper. In the following we respond point-by-point to the comments and suggestions provided.

Reviewer #1:

I have carefully reviewed the manuscript and find it to be a timely, well-executed, and highly valuable contribution to the field of battery materials and recycling economics. The paper addresses a significant and pressing challenge: understanding the cost parity between virgin and recycled materials across a wide array of cathode active materials (CAMs). This is particularly important in the context of growing demand for sustainable, circular, and economically viable battery value chains.

One of the manuscript's strongest features lies in its use of transparent, granular, and high-quality datasets for the production and recycling cost modeling. The authors provide detailed input assumptions across mining, refining, and hydrometallurgical recycling stages. This level of detail is rarely available in the literature and offers an essential foundation for both replication and future research. The regional differentiation, cost allocation methodology, and breakdown by chemistry enhance the practical value of the results.

Crucially, the inclusion of pricing and cost structures supported by public and semi-industrial sources opens the door to a broader use of this model and its assumptions. The modeling framework can be extended or adapted in future work to assess policy impacts, cross-regional competitiveness, or different recycling scenarios. The availability of structured and traceable price data fills a clear gap in the literature and can serve as a benchmark dataset for future techno-economic assessments in the battery space.

In addition, the paper is very well written. The flow of arguments is logical, the methodology is clearly explained, and the figures and supplementary materials are well designed and informative. The authors also make a clear effort to contextualize their results in terms of real-world relevance and future technology trajectories, particularly regarding the increasing role of recycled materials in sustainable battery production.

Although the study focuses on hydrometallurgical recycling, this is clearly stated and well justified. The exclusion of alternative technologies does not diminish the value of the work, given the depth of the analysis provided.

In summary, I strongly recommend this manuscript for publication. It offers excellent methodological rigor, high relevance to both academia and industry, and a uniquely valuable dataset that will undoubtedly serve as a reference point for ongoing and future studies in the field of battery recycling and materials sustainability.

Thank you for this very positive feedback! We deeply appreciate you taking the time to review our paper. Based on your expertise, we are delighted of your comments on the manuscript and of the strong recommendation to publish this manuscript in *Nature Communications* without further improvement. This is an invaluable reward to our work, and we are feeling even more motivated and reinforced to further advance this crucial research field of battery material supply cost and resilience by analysis with this level of detail. Again, thank you!

Reviewer #2:

This study takes a detailed approach to provide cost estimates for primary battery minerals (globally) and recycled minerals (U.S.). The study assesses various configurations of obtaining minerals supply that could reduce total battery costs for the U.S., making them more cost competitive. This is an impressive and detailed study, well-written, with interesting analysis and figures. The topic is highly relevant to the current policy landscape. I would be glad to see this study published in Nature Communications. A few comments to address are below.

Thank you for reviewing our manuscript and for considering this analysis as an ‘impressive’ study of great detail. We much appreciate your feedback and welcome your constructive comments that further improves this manuscript’s value for the community including academics, policymaker, and industry representatives. Furthermore, we want to express our gratefulness for your final recommendation to publish this analysis in Nature Communications.

Comments:

1. The effort you have taken to compile publicly available mine information is commendable. However, are you able to comment on the representativeness of this data, i.e. how well does publicly reported cost information cover the spectrum of costs associated with all mining projects, or those that are included in proprietary data?

Indeed, this aspect is both a major advantage of our method and a potential of future research to refine this analysis by a different scope. As you mentioned correctly, we rely this analysis on a large data set of primary company data on production costs (i.e., capital expenditures, operating expenditures, sustaining capital expenditures, and closure costs) for mining, refining and conversion to battery-grade materials. Most importantly, these data are mainly publicly available because of regulations on information disclosure of publicly traded companies in most regions. This, of course, leads to some data being not available, in particular in production in China and individual important countries, such as the DRC for cobalt or Indonesia in terms of nickel. Although we identified individual cost data here, this can be an area of improvement by adding proprietary data (if available at all). However, we believe that our cost data within the regions are very representative since, first, they cover most of the major production assets in crucial regions (e.g., Tenke Fungurume mine in the DRC or the Sorowako mine in Indonesia), which therefore make up most of average regional supply costs, and, second, most processes in developing countries with less data availability are rather incumbent and mature, which is why costs are likely to be comparable relative to individual major mining and refining concession costs identified in these regions. In the revised manuscript, we extended our statement on this issue in the paragraph from Line 385 to 392.

2. Relatedly, in assessing the cost competitiveness of scenarios with different regional mixes, my question is to what extent we can assume that the average cost used for each region here is truly represent of average costs to meet much larger volume needs, e.g. to meet entire U.S. battery supply chain needs? I have this concern especially because it seems that the costs for several regions are based on only a single facility analyzed in a particular region (from Table 1), and as we see in Figure 2, there could be considerable variation in cost across regions. Moreover, costs may of course scale up over the course of a mine’s lifetime, or change due to other volatilities in the market? To what extent is it a concern to here rely on a single average cost for each region, in making broader assessments about which strategies for regional sourcing could lower U.S. battery costs?

Please also refer to response to the first comment above - We agree that the demand of materials might have an impact on the actual regional supply costs. For example, in theory, the price for commodities is determined by the production cost curves on the market, which encompasses the cumulative demand and the cost profile of all producers (see a comparable presentation in Figure 2). The actual price is therefore determined by the maximum production costs to meet demand. Now, in this manuscript, we assume basically to aspects: First, U.S. demand equals potential supply by a country, and, second, supply costs are determined by a weighted average of production costs in terms of total mine lifetime supply capacity. However, we agree with your comment and acknowledge that costs are averaged and decoupled from supply/demand dynamics driven by current demand levels and future demand developments in the U.S. and globally, as we mention in the discussion section of the revised manuscript. Since this is a very complex

modelling challenge, we assessed this to be out of scope for this study, and motivated to integrate this into future research based on our findings (see Line 392 to 398 now).

3. Figure 7 (and understanding all of the different cases) was a bit confusing for me and took a while to understand. The paragraph preceding this figure could benefit from more detailed description of the different scenarios here and the Figure itself could perhaps use better signposting of the different scenarios.

We appreciate the reviewer pointing this out. We agree that the preceding paragraph to Figure 7 requires refinement to facilitate the understanding of the figure. We now expanded the paragraph and added clarifying information.

Minor comments:

1. Line 254-255: Wording (“surpass”) is a little confusing here. The recycling options costs do “surpass” (are higher than) that of the U.S. primary supply scenario. Maybe change to “The recycling options approach, but do not yet fall below...”? Similar point in Line 279.

Thank you for the suggestion! ‘Fall below’ fits better here, which is why we the wording in the revised manuscript.

Reviewer #3:

The manuscript addresses the production costs of battery materials, particularly cathode active materials, from a U.S. perspective. It compares both primary and secondary material supplies using supply chain scenarios from different countries and regions. The calculations are based on a large sample of mining and processing facilities.

The manuscript covers an interesting and relevant topic. It is well-written and includes comprehensible and informative figures. The supplementary material supports a deeper understanding of the calculations and analyses performed. Some formal improvements are needed, mainly in the reference section. It would also be useful to discuss the impact of U.S. tariffs and their development on the results. Nevertheless, the paper is a valuable addition to the discussion on battery material supply.

Therefore, we recommend that the authors resubmit a revised manuscript that addresses the comments listed below (minor revision).

Major comments

• Throughout the paper, “NMC 95” is used instead of “NMC 955”. If the combination of 90% nickel (Ni), 5% manganese (Mn), and 5% cobalt (Co) is meant, use the more common abbreviation “NMC 955”. Otherwise explain what “NMC 95” means. This applies to many parts of the paper and the supplementary material.

It is indeed NMC 95, which translates to 95% Ni, 2.5% Co, and 2.5% Mn. We base our bill-of-materials for all NMC CAMs on the well-known GREET model. However, we agree that a precise definition of the metal contents has been missing so far, which is why we now added an explanation in the revised manuscript (see Line 69-70 and 427-429 now).

• The abstract mentions “over 80” facilities. In line 79, for example, 79 facilities are mentioned. However, Table 1 and the supplementary Excel sheet list 93 facilities. The difference between 79 and 93 likely stems from facilities that process multiple materials. To increase clarity, the number of facilities examined should be explained once, and the resulting number (79 or 93) should be used throughout the paper and the abstract.

Thank you for your thorough reading! Indeed, this was slightly misleading in the manuscript so far. To clarify: we analyzed primary data of 79 mining and refining facilities (these steps often occur in the same location) and 4 conversion plants for producing batter-grade cobalt sulfate and nickel sulfate from refinery output streams, resulting

in over 80 facilities in total. In the revised manuscript, we now streamlined to precise description of mines, refineries, and conversion plants to avoid confusion with number of facilities in each category. Moreover, we added a note to Table 1 to clarify that the total number of mining and refining facilities exceeds 79 driven by by-products overlaps in cobalt/nickel production facilities.

• The discussion section could be improved by stating the limitations of the calculations in the methods section (e.g., that transportation was not included in the results).

Thanks for the valuable suggestions! We added a note to include transport in future research to the discussion section (see Line 399-400 in revised manuscript).

• As the paper focuses on the U.S., and many of the scenarios include material imports to the U.S., the discussion should address current U.S. tariffs and their development.

Very good point, thank you! In the discussion sections, we now suggest to add the effect of tariffs and export restrictions to future analysis based on our findings (see Line 400-401 in revised manuscript).

• Please carefully revise the reference section, as some references are incomplete (e.g., #23 has no author or year), contain spelling errors (e.g., #7 has “enrgy” instead of “energy”), or have strange abbreviations (e.g., #11 has “Commission, E.” instead of “European Union”). Furthermore, different styles were used (e.g., #24 has no abbreviated first author), and the link in #20 does not work. The same applies to the reference section of the supplementary material.

Thank you for your detailed review! We have edited the formatting of the references to comply with the style guides of Nature Communications, as well as correcting prior spelling errors. We will continue to work with the article production team of Nature Communications on final formats if any other issues remain.

Minor comments

• In line 144, a comma must be used instead of a period after the percentage sign in the parentheses after “DRC”.

• In line 207, “as” should be replaced with “are” (the sixth word).

• In line 360, “modularity” is excessive.

• In the Methods section, some abbreviations are introduced again (e.g., NMC, CAM, EoL). This does not seem necessary.

• In line 432, “LOM” is referred to as “total-lifetime”. On page 2, “LOM” was introduced as “life-of-mine”. Please stick to one term.

Again, thanks for your eye of detail! We have edited these small mistakes in the revised manuscript.